# Hierarchical Approach for Breast Cancer Histopathology Images Classification

Nidhi Ranjan, Pranav Vinod Machingal, Sunil Sri Datta Jammalmadka, Veena Thenkanidiyoor

Department of Computer Science and Engineering
National Institute of Technology Goa
Ponda 401403, Goa, India
nidhi.ranjan145@gmail.com, pranav.machingal6@gmail.com
sunilsreedatta741@gmail.com, veenat@nitgoa.ac.in

A. D. Dileep

School of Computing and Electrical Engineering
Indian Institute of Technology Mandi
Mandi 175001, H.P, India
addileep@iitmandi.ac.in

## Abstract

Detection of cancer and its stages involves analysing histopathology images. Analysing histopathological images is a highly tedious task that requires long attention span. In this paper we propose an approach to detect cancer and its stages by suitably classifying histopathology images. Histopathology images of normal and cancerous tissues exhibit a very complex distinctive geometric structures. A classifier built for detecting cancer and its stages should be able to extract discriminative features. In this paper we propose to adapt CNNs for classification of histopathology images. We propose to build classifiers that use multiple CNNs in hierarchical manner. The multi-CNN based hierarchical classifier is found to be useful. The effectiveness of the proposed CNN-based Hierarchical classifier is studied using the BACH challenge dataset.

## 1  Introduction

Cancer is a paramount disease causing high death rates worldwide. As per International Agency for Research on Cancer (IARC) there are about 8.2 million deaths and are expected to increase by 27 million in 2030. Breast Cancer is second most common cancer in woman, with high mortality rate. Breast tissue histopathology images allow the pathologists to histologically assess the microscopic structure and elements of the tissue which allows to distinguish between normal, non-malignant and malignant tissue. Cancerous tissues may correspond to either InSitu, Invasive or Benign stages of cancer. Analysis of histopathological images is a highly time consuming task and requires experience and good attention span. Early diagnosis of the disease increases the chances of full recovery substantially. This pushes the need for computer-aided quantitative analysis which can produce objective results, and significantly improve the accuracy of diagnosis.

In this work we propose an approach to classify histopathology images of tissue biopsy. The histopathology images being highly complex, we also explore the use of hierarchical classifier which is expected to give better performance. It is important for classifier to capture discriminative features among the classes to be successful in classifying the histopathological images. This requires

1st Conference on Medical Imaging with Deep Learning (MIDL 2018), Amsterdam, The Netherlands.

careful selection of features while building the classifier. Coming up with a good set of hand crafted features is not a trivial task. Recently convolutional neural networks(CNNs) are found to be effective in automatically extracting the necessary set of discriminative features for a given classification task. In this work, we explore the use of CNNs to build a hierarchical classifier for classification of histopathology images corresponding to cancerous tissue. We build a hierarchical classifier of two levels in two ways. One way is to use one CNN each in each level. Second way is to use a binary classifier in first level and a set of binary classifiers in second level. The binary classifiers aim to discriminate between a pair of classes. The efficiency of the proposed multi-CNN based hierarchical classifier is studied using BACH dataset. The important contributions of this work are:

1. adaptation of pre-trained CNN for the classification of histopathology images
2. hierarchical classifiers to address inter-class similarity and intra-class variability issues in histopathology images.
3. usage of multiple CNNs to build a hierarchical classifier.

This paper is organized as follows: Related work is presented in Section 2. In Section 3, proposed approach is presented. Experimental studies is presented in Section 4. In Section 5 we present conclusions.

## 2 Related Work

Medical Image analysis has been an ever advancing field for many years. Automatic processing of images for breast cancer classification has been in the literature for around 40 years with some major groundbreaking works but it still remains a exigent problem because of intricate structure of the microscopy images. Advancements in machine learning & image processing techniques have allowed building of Computer-Aided Detection/Diagnosis (CAD / CADx) systems which can be used to help pathologists in making classification and in turn improve diagnosis. The inborn complexity of histopathological images makes classification of these images into Benign, Invasive, Insitu & Normal classes the primeval goal in building such CAD/CADx systems.

Analysis of histopathology images for cancer classification can be performed either at cell level or at tissue level. For analyzing at cell level it is necessary to first segment out the cell nuclei from histopathology image and then recognize the stage of cancer. At tissue level, the histopathology image of the tissue is taken for analysis. These images express very complex texture and a lot of ambiguity among the classes. Kowal *et al*. [1] compared a set of algorithms for nuclei segmentation and achieved an accuracy of 96% - 100% by majority voting on 10 images each, for for classification into benign or malignant on patient-wise data. Filipczuk *et al*. [2] analyzed cytological images of Fine Needle Biopsy(FNB), where they estimated cell nuclei by using circular Hough transform. Further the detected nuclei are classified using a Support Vector Machine(SVM) based classifier into benign and malignant classes. George *et al*. [3] used a similar approach of nuclei segmentation using Neural Networks. The segmented nuclei are further classified using SVM. Brook *et al*. [4] and Zhang *et al*. [5] worked on three class classification problem by classifying microscopy images into normal, invasive carcinoma and insitu carcinoma. Their results were accomplished on the dataset from Israel Institute of Technology. Zhang *et al*. [5] proposed a method which used a cascaded classification approach with a rejection option. At the first layer, all the images are classified using parallel SVM classifiers. The images that are miss-classified in a certain number of classifiers are considered as difficult cases which are further classified in a second level of cascade where another classification system such as Artificial Neural Networks (ANNs) is used.

Ciresan *et al*. [6]trained a Convolutional Neural Network(CNN) for detection of mitosis in Hematoxylin and eosin(H&E) breast biopsy slides by training on patch sizes of 101 x 101. Teresa *et al*. [7] have used an augmented dataset of the BreakHis dataset [10] after normalizing the images in the training dataset by subtracting the mean from each of the color channels. The dataset was augmented using mirroring and rotation. After normalization and augmentation, patch wise classifiers using a CNN-SVM classifier was built. Fabio *et al*.[8] use a variant of AlexNet [9] and trained it on the BreaKHis database. Small image patches, that of the size of images in Canadian Institute for Advanced Research(CIFAR) dataset [12] were used to learn the parameters of the CNN during training. Since microscopy images contain a huge amount of texture & detail the surmise was that the patches extracted possibly contained information that could aptly help to train the model. Experiments were conducted on patch size of 32 x 32, 64 x 64, with & without sliding window. The

classification of the whole image is done using different fusion rules common in literature such as Sum rule or Product rule and Max rule.

In this work, we propose to build a CNN-based classifer for classification of histopathology images into Normal, Benign, Invasive and Insitu classes. We also propose to explore CNN-based hierarchical classifiers.

# 3 Classification of Breast Cancer Histopathology images

Classification of Breast Cancer histopathology images is a very complex task due to the complex and diverse structures found in these images. These images have rich geometrical structure which make it difficult to learn the discriminative features among the classes. This paper is written as part of the BACH Challenge 2018. The dataset used is the one provided with the challenge. A few images from the 4 classes of the dataset are shown in Figure 1. It is seen from Figure 1 that the images of a class exhibit strong variability. It is also seen that there exists strong inter class similarity among the classes. A classifier should be able to capture the discriminative features among the classes in such complex classification tasks. Convolutional Neural Networks(CNN) are shown to be effective in building discriminative classifiers. We propose to tune a pretrained AlexNet to the task of classification of histopathology images.

In this work we propose to build CNN-based classifiers using AlexNet [9] trained on 1000 classes of ImageNet through transfer learning. We first propose a CNN-based classifier for classification of histopathology images. We propose to change the output layer of a pretrained AlexNet model to have only 4 neurons. We explore two approaches for training this network. In the first approach the feature maps of convolutional layers of pretrained Alexnet are retained as it is and only the fully connected layer is retrained. In the second approach, the entire network is retrained. It is seen from Figure 1 that histopathology images of Normal class has distinct appearance than the images of the other 3 classes. This indicates a hierarchical approach to the classification may be effective.

We propose a CNN-based hierarchical classifier as shown in Figure 2. In the first level of the hierarchy we build a CNN-based classifier to discriminate images of Normal class from images of the rest of the classes. In the second level of hierarchy we propose to build three CNN-based binary classifiers. Each binary classifier is built to discriminate one class from another. For example, a binary classifier to discriminate examples of Benign class from examples of InSitu class. Since there are three classes, we propose to built three such binary classifiers as shown in Figure 2. Any test image that got classified as Rest in the first stage is passed through the three binary classifiers. Each classifier will assign the example to one of the two classes. The test example is assigned to a class with maximum votes. The one-vs-one CNN-based classifiers are built by taking the training data of the respective two classes.

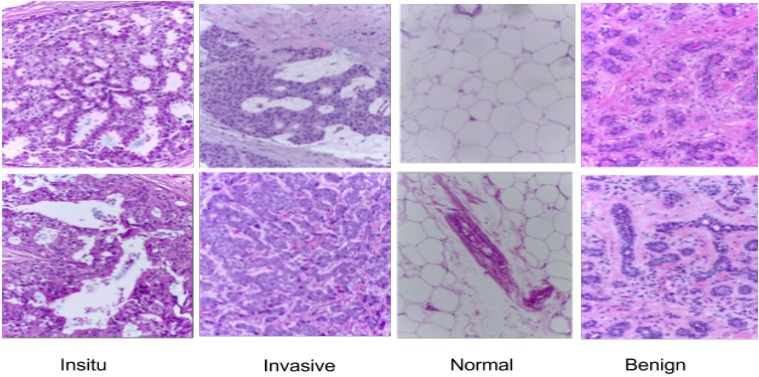

Figure 1: Example images of the four classes found in the Dataset of BACH Challenge. Here, each column corresponds to a class

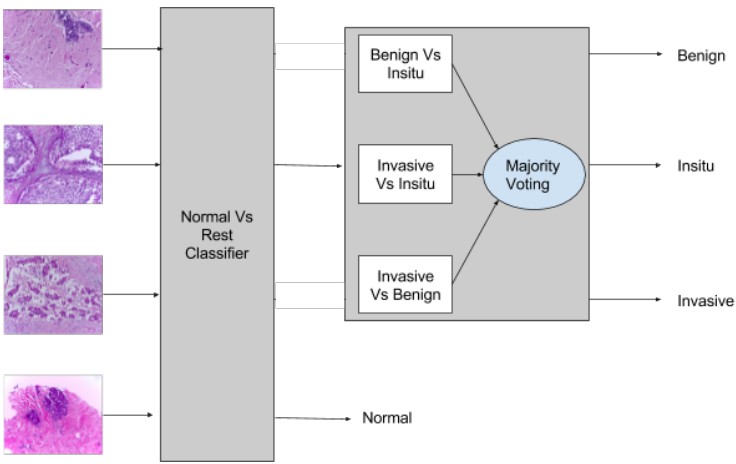

Figure 2: A CNN-based Hierarchical classifier for classification of histopathology images

|  | Classification Accuracy in % on Validation Data |
| --- | --- |
| Retraining only the fully connected layer | 77 |
| Retraining the entire network | 83 |

Table 1: Comparison of classification accuracy of CNN based classifiers built by 1) Retraining only the fully connected layer and 2) Retraining the entire network

## 4 Experimental Studies on Breast Cancer Classification

The studies on breast cancer classification are done on BreaAst Cancer histopathology Images(BACH) dataset of ICIAR Grand Challenge. The dataset comprises of 400 images of 4 classes namely, Benign, Insitu, Invasive and Normal. There are a total of 100 images in every class. We consider 75 images per class for training and 25 images as test set. Each image is of 2048 x 1560 pixels in size. In this work we consider Convolutional Neural Network(CNN) based classifiers for classification of histopathology images. In this work we explored the use of AlexNet [9] for building the classifiers for the classification of histopathology images. The reason for choosing AlexNet is that it is a well established architecture and has shown good performance when used for many tasks involving natural images as well as medical images. AlexNet comprises of five convolutional layers followed by three fully connected layers and also the max-pooling layers.

The studies were implemented using pytorch on 40 CPU cores with 128GB RAM. The learning rate for training was set to 0.001 with batch size of 32 and optimization technique used is Stochastic Gradient descent with a momentum of 0.9.

### 4.1 Studies on CNN-based Classification of Histopathology Images

To build CNN-based classifier for histopathology images we replaced the 1000 nodes in output layer of AlexNet with 4 nodes. We used two approaches for training this CNN for histopathology image classification. The first approach is to retrain only the fully connected layer keeping the feature maps of the convolutional layers of the pretrained model. The second approach is to retrain the entire network considering the feature maps of the convolutional layers of the pretrained models as initial weights. The training is performed using cross entropy loss function. The performance of the CNN-based classifiers built using these two approaches is given in the Table 1. It is seen from Table 1 that the CNN-based classifier where the entire network was retrained is found to perform better than the one which involves retraining only the fully connected layer. In this work , we consider CNN-based classifiers in which the entire network is retrained. The confusion matrix for the CNN-based classifier that involved retraining the whole network is given in Figure 3. Here each class has 25 test examples.

It is seen from Figure 3 that the histopathology images of Normal class are getting correctly classified. It is also seen that the histopathology images of the other 3 classes are getting miss-classified

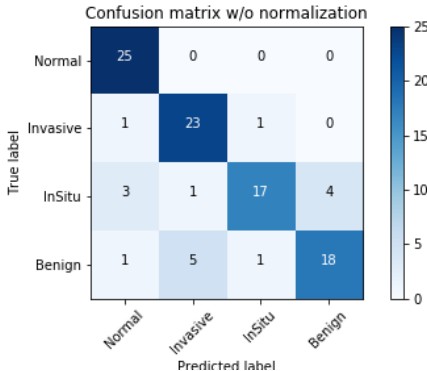

Figure 3: Confusion Matrix obtained by CNN-based classifier by adapting AlexNet

|  | Classification Accuracy in % |
|---|---|
| Hierarchical Classifier 1 : CNN classifier for 3 classes in second level | 79 |
| Hierarchical Classifier 2 : Majority voting based classifier in second level that uses 3 one-vs-one classifiers | 95 |

Table 2: Comparison of classification accuracy of CNN-based Hierarchical classifiers

among themselves. This shows that there exists a lot of inter-class similarity between the Invasive, InSitu, and Benign classes. To improve the classification accuracy, we need to capture effective discriminative features among them. For this we study CNN-based hierarchical classifier in the next section.

## 4.2 Studies on Classification of Histopathology Images Using CNN-based Hierarchical Classifier

In the first level of hierarchy, we build a CNN-based classifier to classify the histopathology images into Normal and Rest classes. The images of Benign, InSitu and Invasive classes are considered as examples for Rest class. Here, the nodes of output layer of AlexNet is replaced with two neurons and the entire network is retrained. In this first level the network is trained to discriminate between histopathology images of Normal class from the images of the rest of the classes. In the second level of hierarchy we first build a CNN-based classifier by considering 3 nodes in the output layer and retraining the entire network. This CNN is retrained using data of only these 3 classes. We also built the second level of hierarchical classifier by building 3 one-vs-one CNN-based classifiers and using the majority voting decision logic. The 3 one-vs-one classifiers built are Benign vs InStiu, Invasive vs InSitu and Invasive vs Benign. A one-vs-one CNN-based classifier is built by considering only the data of those 2 classes and having 2 neurons in the output layer. Basically, these CNN-based classifiers learn to discriminate between the 2 classes considered. Since only two classes are considered there are only 150 images are available for training. It is impossible to retrain a CNN using only these many images. To make the retraining of the model possible, the dataset was augmented with flips and rotations to get 600 images per class. The accuracies of the two proposed hierarchical classifiers is compared in Table 2.

It is seen from Table 2 and Table 1 that the hierarchical classifiers are performing better than CNN-based classifiers that involved 4 output layer neuron that involved retraining of only the fully connected layer. It is also seen that using one-vs-one CNN-based classifiers in second level of hierarchy is found to perform better than a 3 class CNN-based classifier in second level. In Figure 4, we give the confusion matrix for the Hierarchical Classifier 2. From Figure 3 and 4 it is seen that the classification of histopathology images of Invasive , Insitu and Benign classes is significantly improved when hierarchical classifier is used. It is also seen that the inter class confusions among Insitu, Benign and Invasive classes is significantly reduced when the Hierarchical Classifier 2 is used.

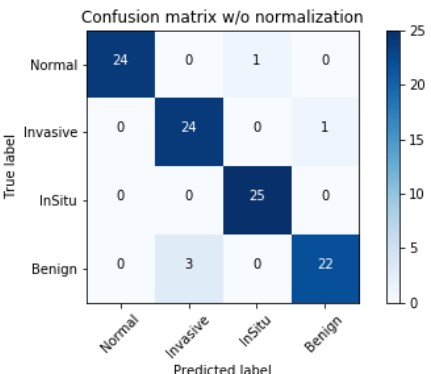

Figure 4: Confusion Matrix for Hierarchical Classifier 2

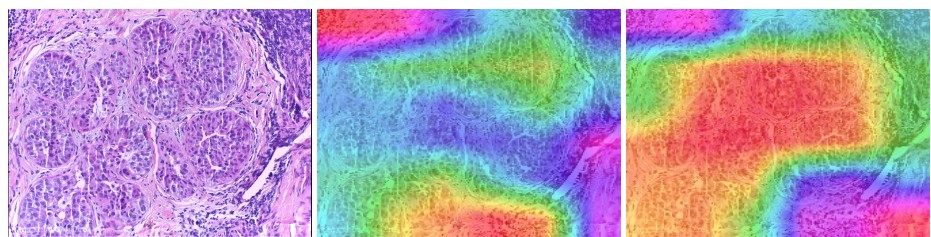

Figure 5: An image from In- Figure 6: Grad-CAM Figure 7: Grad-CAM
vasive class Heatmap for the image Heatmap for the image
in Figure 5 in a 4 class in Figure 5 in a 2 class
CNN(red regions represent CNN built for Invasive and
features network learns to InSitu classes(red regions
attribute to true label) represent features network
learns to attribute to true
label)

In Figure 6 we show the gradient based class activation heatmap for a 4-class CNN for an image from Invasive class shown in Figure 5. In Figure 7 we give the gradient based class activation heatmap for 2 class CNN built for Invasive and InSitu classes. It is seen from Figure 5, 6 and 7 that per class representation of features for classification was more when the model was trained to discriminate between a pair of classes. This is a possible reason for the better classification accuracy of Hierarchical Classifier 2.

## 5 Conclusion

In this paper, a CNN-based hierarchical classifier to classify histopathology images into 4 classes namely, Normal, Benign, InSitu and Invasive is proposed. In the first level of the hierarchical classifier, a two class CNN-based classifier is built to discriminate images of Normal class from the images of the rest of the classes. In the second stage of hierarchy we built a majority voting based classifier where the votes are given by the 3 binary CNN-based classifiers. The binary CNN-based classifiers are built in one-vs-one way to discriminate one class from another. The studies were performed on the dataset provided in the BACH challenge. The proposed hierarchical classifier is found to perform good in classifying histopathology images into Normal, Invasive, Benign and InSitu classes. In future, the proposed approach may be extended to classification of other histopathology images.

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
