# OpenReview forum: "Hierarchical Approach for Breast Cancer Histopathology Images Classification"
_MIDL.amsterdam/2018/Conference — Submitted to MIDL 2018_

### Review · AnonReviewer3 · 2018-05-04
**paper well presented, simple pipeline, weak experiments**

**Rating:** 2
**Confidence:** 2

**Review:**

This paper addresses the problem of breast cancer histopathology image classification by means of hierarchical CNN classifiers. The model build is based on the well known AlexNet architecture.

pros
+ important problem
+ paper well presented
+ simple pipeline

cons
- model could be updated with a more recent (and hopefully better performing) one
- experimental evaluation is not compelling

The paper is well presented and easy to follow. My main concerns are related to the validation of the method.
* Why use AlexNet instead of more recent architectures such as ResNets, DenseNets or Inception?
* The idea of using a pre-trained model (or pre-trained AlexNet), and fine-tuning it by changing the number of outputs to perform a different task has been largely studied in both the computer vision and medical imaging literature. However, the authors propose this as a novel pipeline. Although the application might be novel, previous work should be properly acknowledged and discussed.
* Given the size of the dataset, it would be appropriate to report results on a 5 or 10 fold cross-validation, reporting mean and std.
* In addition to reporting accuracy and confusion matrices, it might be interesting to report AUC. It would also be easier to compare results if tables and confusion matrices of different settings were merged or placed next to each other.
* For binary classification architectures, only one output is needed.
* It seems that data augmentation is only applied in the hierarchical case, to train one-vs-one models. This makes it hard to compare the obtained results with other architectures. For fair comparison, all models should be trained using data augmentation.
* In Figure 7, it seems arbitrary to show the heat map for the invasive vs insitu classes, given that multiple models are trained.
* A couple of papers that might be worth discussing:
https://www.biorxiv.org/content/biorxiv/early/2018/01/04/242818.full.pdf
https://www.biorxiv.org/content/biorxiv/early/2018/02/05/259911.full.pdf

**Special Issue:**

No

---

### Review · AnonReviewer2 · 2018-05-05
**using a cascade of multiple binary classifiers to solve a breast caner histopathology image multi-class classification problem**

**Rating:** 3
**Confidence:** 3

**Review:**

Instead of learning a single multi-class classification neural network, the paper proposed to learn a cascade of multiple binary classifiers. The first binary classifier separates normal vs the rest, and then three binary classifiers are learned to separate benign, in-situ, and invasive. Experimental results on BACH challenge show the effectiveness of the proposed algorithm.

A traditional way of converting a multi-class classification problem is to train a series of all possible pair-wise binary classifiers. It is necessary to compare the proposed algorithm with it.

Another issue regards why the first binary classifier learns to separate "normal" from the rest.

**Special Issue:**

No

---

### Review · AnonReviewer1 · 2018-05-17

**Rating:** 1
**Confidence:** 3

**Review:**



**Special Issue:**

No

---

### Decision · Program_Chairs · 2018-05-15
**Paper47 Acceptance Decision**

Reject